# SPARSECACHE: EXTREME SPARSE CODING FOR KV CACHE COMPRESSION

## ABSTRACT

The growing memory footprint of the Key-Value (KV) cache is a critical bottleneck in Large Language Models (LLMs), significantly hindering inference efficiency. While emerging "precomputed RAG" paradigms promise to reduce latency by precomputing KV caches for entire corpora, their prohibitive storage requirements render them impractical. This paper introduces **SparseCache**, a novel KV Cache compression framework that addresses this bottleneck. *SparseCache* employs an end-to-end learning framework, inspired by the K-SVD algorithm's alternating optimization, to create separate globally shared dictionaries for Key and Value vectors across the model. By optimizing these dictionaries directly against a reconstruction loss objective, *SparseCache* captures fundamental KV Cache redundancies more holistically than prior per-layer methods. Extensive experiments show that *SparseCache* achieves a state-of-the-art compression ratio of up to $17.7\times$ while preserving model accuracy on challenging long-context benchmarks. Notably, it maintains high performance at over $8\times$ compression, a level where competing techniques struggle. By enabling high-fidelity compression, *SparseCache* makes the "precomputed RAG" paradigm practical and feasible, leading to reduced Time-To-First-Token (TTFT) and improved overall system throughput.

## 1 INTRODUCTION

Large Language Models (LLMs) face a critical bottleneck in long-context applications: the Key-Value (KV) cache. This cache, essential for efficient autoregressive generation, stores intermediate attention states but grows linearly with sequence length, consuming vast amounts of memory and causing high inference latency (Li et al., 2024; Yang et al., 2024; Adnan et al., 2024; Brandon et al., 2024). This problem is particularly acute in Retrieval-Augmented Generation (RAG) systems, where concatenating retrieved documents with a query creates long input contexts. The standard RAG workflow requires a computationally intensive *prefill* step to process this entire context at once, severely degrading the Time-To-First-Token (TTFT) and hindering real-time interaction (Jin et al., 2024; Lu et al., 2024; Lee et al., 2025; Ma et al., 2024).

To combat this latency, a "precomputed RAG" paradigm has emerged that moves the expensive KV Cache generation for corpus documents offline (Lu et al., 2024; Ma et al., 2024; Jin et al., 2024; Yao et al., 2025). While this approach drastically reduces online computation, it introduces a prohibitive new challenge: a massive storage bottleneck. Storing the full, uncompressed 16bit KV Caches for an entire document corpus is often infeasible, rendering this promising low-latency paradigm impractical at scale (Jin et al., 2024; Lee et al., 2025).

This paper introduces **SparseCache**, a novel dictionary-based compression framework designed to make the "precomputed RAG" paradigm practical and scalable. As illustrated in Figure 1, *SparseCache* compresses the vast set of precomputed KV Caches into a highly compact collection of sparse codes. During online inference (Figure 1, right), the system retrieves these lightweight codes, decompresses them on-the-fly into high-fidelity KV-states, and seamlessly integrates them with the query's cache. This methodology eliminates the costly prefill step of standard RAG while circumventing the storage crisis of naive precomputation, thereby achieving low latency with a manageable memory footprint.

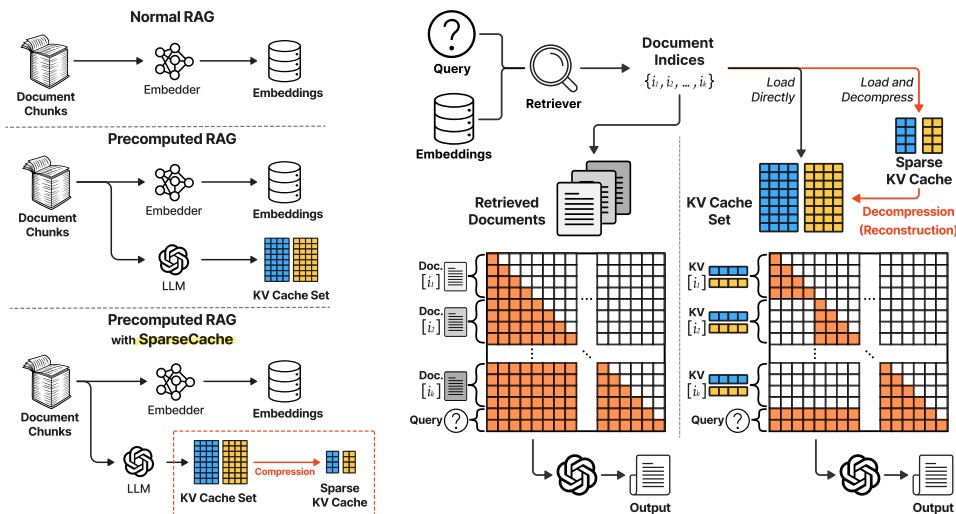

Figure 1: Comparison of RAG workflows across offline preparation (left) and online inference (right). **(Left)** 'Precomputed RAG' creates a massive KV Cache Set, causing a storage bottleneck. *SparseCache* resolves this by compressing the set into a manageable *Sparse KV Cache*. **(Right)** 'Normal RAG' suffers from high Time-To-First-Token (TTFT) due to a costly prefill step. *SparseCache* eliminates prefilling by loading and reconstructing its lightweight caches, significantly reducing memory footprint and improving TTFT.

The core innovation of *SparseCache* is an end-to-end framework for learning global Key and Value dictionaries, inspired by the alternating optimization structure of the K-SVD algorithm (Aharon et al., 2006). Unlike methods that learn separate dictionaries for each layer, SparseCache learns a single, unified dictionary for all Key vectors and another for all Value vectors across the entire model. This global approach captures the most fundamental and redundant structures within the KV Cache space more holistically and compactly. The term *end-to-end* signifies our approach of employing a deep-learning-style training paradigm using external text datasets to learn these dictionaries, rather than traditional Orthogonal Matching Pursuit (OMP)-based dictionary learning. By primarily optimizing these dictionaries against reconstruction loss, *SparseCache* achieves an intelligent form of lossy compression that preserves model accuracy at high compression ratios, as minimizing reconstruction error directly correlates with maintaining the LLM's performance objectives on downstream tasks.

Our primary contributions are:

- We propose *SparseCache*, a novel end-to-end framework that learns global K-SVD-inspired dictionaries to enable scalable and practical "precomputed RAG" systems.
- We demonstrate that *SparseCache* significantly outperforms state-of-the-art compression methods in terms of compression ratio, model accuracy, and inference speed (TTFT).
- We provide in-depth analysis of the learned dictionaries and the factors influencing *SparseCache*'s performance, offering new insights into the structure of LLM KV Caches.

## 2 LITERATURE REVIEW

The significant memory and latency overhead of the Key-Value (KV) cache has spurred a wide range of compression strategies for LLM inference (Li et al., 2024; Gao et al., 2025; Liu et al., 2025). These methods can be broadly categorized into three main approaches: pruning, quantization, and low-rank decomposition. Pruning techniques selectively discard less important KV pairs based on metrics like attention scores (Zhang et al., 2023; Liu et al., 2023; Adnan et al., 2024; He et al., 2024; Ni et al., 2025), token position (Xiao et al., 2023; Ge et al., 2023), or recency in a sliding window (Zuo et al., 2025; Xiao et al., 2023), though they risk the irreversible loss of long-range information

(Yang et al., 2024; Liu et al., 2025). Quantization methods reduce the numerical precision of KV Cache entries, with advanced techniques focusing on adaptive bit-widths (He et al., 2024; Yang et al., 2024; Zhang et al., 2024) and specialized data types to handle the unique distributions of Keys and Values (Hooper et al., 2024; Liu et al., 2024b; Tao et al., 2024; Zhang et al., 2024). Lastly, low-rank decomposition methods exploit structural redundancy by applying techniques like Singular Value Decomposition (SVD) to individual or grouped layers to find shared subspaces (Chang et al., 2025; Saxena et al., 2024; Chang et al.; Yankun et al., 2025).

A distinct and emerging paradigm is dictionary learning, which represents KV vectors as a sparse linear combination of "atoms" from an overcomplete dictionary, a principle established by algorithms like K-SVD (Aharon et al., 2006; Kim et al., 2024; Zhang et al., 2025). In the context of KV Cache compression, Lexico (Kim et al., 2024) pioneered this approach by learning a separate, universal dictionary for each layer of the model. While effective, this design choice results in managing numerous dictionaries, which proportionally increases the storage and complexity overhead with the model's depth. For massive "precomputed RAG" systems, where compressed caches for an entire corpus must be stored, this overhead can become a significant bottleneck.

*SparseCache* diverges fundamentally from this design. We propose learning a single, *globally* shared dictionary for all Key vectors and another for all Value vectors across the entire model. This global approach is designed to capture the most fundamental and redundant structures of the KV Cache space more holistically and compactly, thereby dramatically reducing the storage footprint. This key difference makes *SparseCache* an especially suitable solution for enabling the "precomputed RAG" paradigm (Lu et al., 2024; Ma et al., 2024; Jin et al., 2024; Yao et al., 2025; Lee et al., 2025).

Furthermore, a critical distinction lies in the handling of multi-head attention (MHA) KV Caches. Whereas Lexico compresses each head's vector individually, *SparseCache* concatenates all head vectors for a given token into a single, higher-dimensional signal before compression (Kim et al., 2024). While this approach increases the computational cost of Orthogonal Matching Pursuit (OMP) for the offline precomputation, our empirical results demonstrate it yields a superior overall trade-off between compression and model performance. Crucially, in a "precomputed RAG" paradigm, this intensive compression step has no impact on online inference latency.

## 3 PRELIMINARIES

This section briefly outlines dictionary learning and the K-SVD algorithm, which are foundational to the *SparseCache* framework. The goal of dictionary learning is to represent a set of signal vectors, $\mathbf{Y} = [\mathbf{y}_1, \ldots, \mathbf{y}_p] \in \mathbb{R}^{m \times p}$ (in our case, KV vectors), as a sparse linear combination of "atoms" from an overcomplete dictionary $\mathbf{D} \in \mathbb{R}^{m \times n}$. This is achieved by finding both $\mathbf{D}$ and a sparse coefficient matrix $\mathbf{X} \in \mathbb{R}^{n \times p}$ that solve the following optimization problem:

$$\min_{\mathbf{D}, \mathbf{X}} \|\mathbf{Y} - \mathbf{DX}\|_F^2 \quad \text{s.t.} \quad \forall i, \|\mathbf{x}_i\|_0 \leq s \tag{1}$$

where $\|\mathbf{x}_i\|_0$ is the number of non-zero elements and $s$ is the desired sparsity.

K-SVD (Aharon et al., 2006) is a widely-used iterative algorithm that solves this by alternating between two stages:

**1. Sparse Coding Stage:** With the dictionary $\mathbf{D}$ fixed, this stage finds the optimal sparse representation $\mathbf{x}_i$ for each signal $\mathbf{y}_i$. Since finding the exact $L_0$-sparse solution is NP-hard, greedy methods like Orthogonal Matching Pursuit (OMP) (Tropp & Gilbert, 2007) are employed to find an approximate solution.

**2. Dictionary Update Stage:** With the sparse codes $\mathbf{X}$ fixed, the dictionary atoms $\mathbf{d}_k$ are updated one by one. For each atom, K-SVD identifies the signals that use it and updates the atom (and its corresponding coefficients) to better fit these signals, typically via Singular Value Decomposition (SVD).

This alternating optimization structure, which progressively refines both the dictionary and the sparse codes, is a core principle that inspires the *SparseCache* framework. A more detailed mathematical formulation of the classic K-SVD update stage is provided in Appendix.

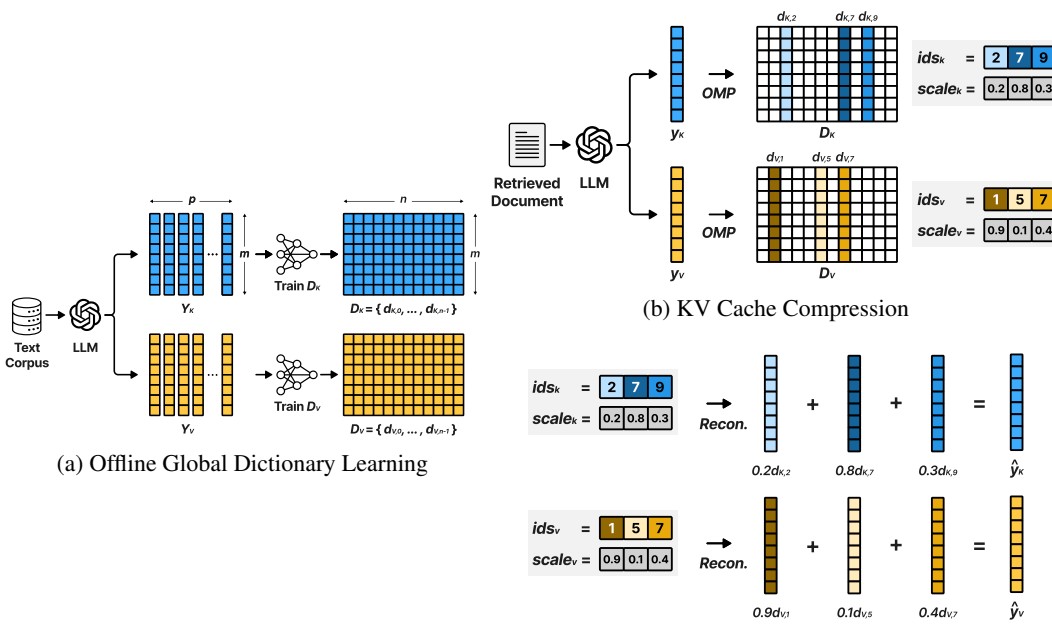

(b) KV Cache Compression

(a) Offline Global Dictionary Learning

(c) KV Cache Decompression (Reconstruction)

Figure 2: Overview of the *SparseCache* framework. (a) Global dictionaries $\mathbf{D}_K$ and $\mathbf{D}_V$ are learned offline from a training set of KV Cache data. (b) Corpus documents are processed, their KV Caches (composed of individual Key vectors, $\mathbf{y}_K$, and Value vectors, $\mathbf{y}_V$) extracted, and then compressed using the learned dictionaries and OMP. This stores sparse codes, which consist of the indices of the chosen dictionary atoms and their corresponding coefficient values (e.g., $(\mathbf{ids}_k, \mathbf{scale}_k)$ in the diagram for a single Key vector, which correspond to the sparse code $(\mathbf{ids}, \mathbf{scale})$ in Algorithm 2). (c) During inference, these sparse codes for retrieved documents are used with $\mathbf{D}_K, \mathbf{D}_V$ to reconstruct the original KV vectors (denoted $\hat{\mathbf{y}}_K, \hat{\mathbf{y}}_V$) for the LLM.

## 4 METHODS

*SparseCache* adapts dictionary learning, using Orthogonal Matching Pursuit (OMP), for compressing and managing KV Caches in Retrieval Augmented Generation (RAG) systems. Unlike typical LLM inference where KV Caches are transient, in a RAG context, KV Caches from corpus documents can be precomputed and stored in a compressed form. *SparseCache* learns global dictionaries, $\mathbf{D}_K$ for Key vectors and $\mathbf{D}_V$ for Value vectors, derived from an LLM's KV Cache activations on a training dataset. These dictionaries are shared across all layers and heads. The overall *SparseCache* process, illustrated in Figure 2, encompasses three main stages: (a) offline global dictionary learning, (b) KV Cache compression for RAG corpus preprocessing, and (c) KV Cache decompression during RAG query processing. The pseudocode for these stages is provided in Algorithms 1, 2, and 3 in the Appendix.

GLOBAL DICTIONARY LEARNING

The foundation of *SparseCache* is the offline learning of global dictionaries, as depicted in Figure 2a. This process is iterative, aiming to minimize reconstruction error.

### 4.0.1 INPUT DATA AND DICTIONARY DESIGN

Key vectors (e.g., $\mathbf{y}_{i,K}$) and Value vectors (e.g., $\mathbf{y}_{i,V}$) are precollected from an LLM's attention layers by processing a training dataset. For our experiments, we used a 3M token subset sampled from the Wikitext dataset. These form the training matrices $\mathbf{Y}_K^{\text{train}}$ and $\mathbf{Y}_V^{\text{train}}$, respectively:

$$\mathbf{Y}_K^{\text{train}} = [\mathbf{y}_{1,K}, \mathbf{y}_{2,K}, \dots, \mathbf{y}_{p,K}] \tag{2}$$

$$\mathbf{Y}_V^{\text{train}} = [\mathbf{y}_{1,V}, \mathbf{y}_{2,V}, \dots, \mathbf{y}_{p,V}] \tag{3}$$

Separate global dictionaries $\mathbf{D}_K$ and $\mathbf{D}_V$ are learned. A sparsity target $s$ defines the desired number of non-zero coefficients per vector.

### 4.0.2 TRAINING PROCESS

The training process adopts the alternating optimization structure of K-SVD. However, unlike traditional K-SVD or OMP-based dictionary learning methods, *SparseCache* employs a deep-learning-style training paradigm using external text datasets. This enables an "end-to-end" approach to learning dictionaries, where the training process benefits from standard deep learning optimization techniques.

In each iteration, the process begins with the sparse coding stage. For mini-batches of Key vectors ($\mathbf{Y}_{K,\text{batch}}$) and Value vectors ($\mathbf{Y}_{V,\text{batch}}$) sampled from the training sets, we employ Orthogonal Matching Pursuit (OMP) to compute the optimal sparse coefficient matrices, $\mathbf{X}_{K,\text{batch}}$ and $\mathbf{X}_{V,\text{batch}}$, using the current dictionaries $\mathbf{D}_K$ and $\mathbf{D}_V$.

Next, in the dictionary update stage, we deviate from the traditional K-SVD's atom-by-atom SVD update. To handle the massive volume of KV vectors effectively and leverage modern GPU architectectures, *SparseCache* adopts a simultaneous update rule based on mini-batch gradient descent, akin to training a neural network. The entire dictionaries $\mathbf{D}_K$ and $\mathbf{D}_V$ are updated to minimize their respective reconstruction losses, $\mathcal{L}_K$ and $\mathcal{L}_V$:

$$\mathcal{L}_K = \|\mathbf{Y}_{K,\text{batch}} - \mathbf{D}_K \mathbf{X}_{K,\text{batch}}\|_F^2 \tag{4}$$

$$\mathcal{L}_V = \|\mathbf{Y}_{V,\text{batch}} - \mathbf{D}_V \mathbf{X}_{V,\text{batch}}\|_F^2 . \tag{5}$$

The primary objective of learning dictionaries with reconstruction loss is to ensure that the lossy compressed KV Cache closely approximates the lossless baseline KV Cache. This direct fidelity is crucial because, in lossy compression, better reconstruction quality inherently translates to superior performance on LLM benchmarks by minimizing the degradation introduced by compression. Thus, optimizing reconstruction loss serves as the direct proxy for achieving high LLM performance.

This approach allows for efficient, large-batch training. After each gradient step, all dictionary atoms are re-normalized to unit $L_2$-norm to ensure training stability. This iterative refinement of sparse codes and dictionary atoms continues until convergence, specifically when the reconstruction error stabilizes.

### 4.1 KV CACHE COMPRESSION FOR RAG CORPUS PREPROCESSING

Once the global dictionaries $\mathbf{D}_K^*$ and $\mathbf{D}_V^*$ are learned, they are used to preprocess the documents in the RAG corpus (Figure 2b). For each document (or document chunk), its KV Cache vectors (individual Key vectors $\mathbf{y}_{j,K}$ and Value vectors $\mathbf{y}_{j,V}$) are compressed. This is an offline step. For each vector, the OMP algorithm identifies a sparse representation. The compressed output, which we refer to as a **sparse code**, consists of two components: the indices of the $s$ selected dictionary atoms and their corresponding non-zero coefficient values. Specifically, for a given vector $\mathbf{y}$, the compression yields a tuple (**ids**, **scale**), where **ids** contains the integer indices of the selected atoms from the dictionary $\mathbf{D}$, and **scale** contains the corresponding floating-point coefficients.

To further enhance storage efficiency, we apply an additional index compression technique. Since our dictionary contains exactly 8192 atoms, each index requires only 13 bits rather than the standard 16 bits used by int16. We implement a bit-packing algorithm (detailed in Appendix A.6) that compresses these indices from 16 to 13 bits, achieving an additional 1.23× compression for the index component. This optimization, combined with the inherent sparsity of our representation, contributes significantly to our overall compression ratios. These codes (e.g., $\mathcal{C}_{doc,K}, \mathcal{C}_{doc,V}$ for an entire document) are then stored in an indexed structure.

### 4.2 KV CACHE DECOMPRESSION FOR RAG QUERY PROCESSING

During RAG query time, relevant document chunks are retrieved from the preprocessed corpus. The corresponding sparse codes, which are a subset of the full compressed corpus $\mathcal{C}_{KV}$, are fetched. This retrieved set of codes, denoted $\mathcal{C}_{\text{retrieved}}$, must then be decompressed. Decompression involves using the stored sparse codes. For each code, the approximate original dense KV vector ($\hat{\mathbf{y}}_{j,K}$ or $\hat{\mathbf{y}}_{j,V}$)

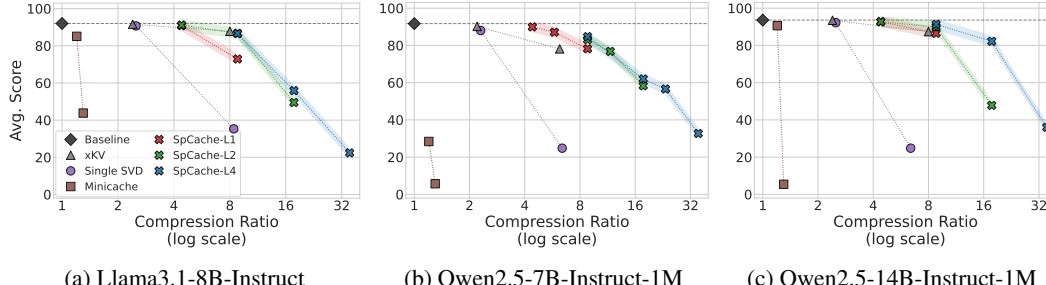

| | |
|---|---|
| (a) Llama3.1-8B-Instruct | (b) Qwen2.5-7B-Instruct-1M |
| | (c) Qwen2.5-14B-Instruct-1M |

Figure 3: Average score vs. Compression ratio across multiple models. The y-axis shows the average score on the *RULER* benchmark, while the x-axis (log scale) represents the KV Cache compression ratio. Across all tested models, *SparseCache* variants (configured by layer sharing $L$ and sparsity $s$) consistently demonstrate a superior trade-off, maintaining high performance at significantly higher compression ratios compared to baselines.

is reconstructed by calculating a weighted sum of the dictionary atoms, where the indices (**ids**) select the atoms and the coefficients (**scale**) provide the weights, as shown in Figure 2c. These reconstructed vectors, along with the query's own KV vectors, are used by the LLM's attention mechanism to generate the final response.

## 5 EXPERIMENTS

This section details the experimental evaluation of the *SparseCache* framework. It begins by outlining the Experimental Setup, including the *RULER* benchmark, the specific LLMs tested (Llama3.1, Qwen2.5), the various baseline methods it's compared against (like *Minicache, SVD, xKV*), and the performance metrics used (*RULER* score and compression ratio). The Main Results and Analysis then present *SparseCache*'s overall superior performance-compression trade-off and its robust, graceful degradation compared to baselines, especially in high-compression scenarios. Following this, the Analysis of *SparseCache* Hyperparameters explains how internal parameters like sparsity and layer sharing can be adjusted to fine-tune the balance between compression and accuracy. The discussion then moves to implications for system performance, showing how *SparseCache* reduces Time-To-First-Token (TTFT) by enabling faster KV Cache loading, which is crucial for long contexts, large models, and distributed systems. Finally, the Memory Footprint analysis quantifies the substantial memory savings achieved by *SparseCache*, demonstrating its practical utility in resource-constrained environments.

### 5.1 EXPERIMENTAL SETUP

We conduct our experiments on the *RULER* benchmark (Hsieh et al., 2024), a standard for long-context evaluation, using a 64K token context to simulate a demanding Retrieval-Augmented Generation (RAG) scenario. Our evaluation includes several models: *Llama3.1-8B-Instruct*, *Qwen2.5-7B-Instruct-1M*, and *Qwen2.5-14B-Instruct-1M*. We compare our approach against multiple baselines, including the uncompressed BF16 baseline, *Minicache* (Liu et al., 2024a) for local redundancy, single SVD compression applied per layer, and *xKV* (Chang et al., 2025), which groups layers before applying SVD. We test several configurations of our method, **SparseCache**, denoted as *SparseCache-L-s*, where $L$ represents the number of consecutive layers sharing a dictionary (e.g., $L1$ for per-layer, $L4$ for sharing across 4 layers) and $s$ indicates the sparsity (the number of atoms per vector). All *SparseCache* configurations use a fixed dictionary size 'n' of 8192. Performance is assessed using two primary metrics: the average *RULER* score, to measure preserved reasoning ability, and the compression ratio, calculated as the original cache size divided by the compressed size.

## 5.2 MAIN RESULTS AND ANALYSIS

Our experimental results, summarized in Figure 3 and detailed in Table 2 in the Appendix, clearly demonstrate the superiority and robustness of the *SparseCache* framework across multiple LLMs of varying architectures and sizes.

As shown in Figure 3, *SparseCache* variants consistently define a superior Pareto frontier compared to all baseline methods in all models tested. Although methods like *xKV* perform well at low compression ratios, *SparseCache* achieves similar or better performance at significantly higher compression levels. For example, on the *Qwen2.5-7B* model (Figure 3b), *xKV-4* achieves an average score of 90.2% at a compression ratio of 2.2×. Remarkably, our *SparseCache-L1-s64* method reaches a nearly identical score of 89.9% but at a compression ratio of 4.4×, doubling memory savings. This trend holds for other models; on *Llama3.1-8B* (Figure 3a), *SparseCache-L2-s256* achieves 91.2 precision at 4.4× compression, comparable to *xKV-4*'s 91.6% at only 2.4× compression.

A key advantage of *SparseCache*, observed in all models, is its graceful degradation in performance as compression increases. This contrasts sharply with competing methods. *Minicache* suffers a catastrophic performance collapse even at minimal compression ratios, and *Single SVD* also exhibits a sharp performance cliff, rendering it unreliable for moderate to high compression. For example, on the *Qwen2.5-14B* model (Figure 3c), *Single SVD* performance drops from 92.2% at 2.5× to just 24.8% at 6.4× compression. *SparseCache*, however, maintains high fidelity over a much wider range. On the same 14B model, at an 8.8× compression ratio, *SparseCache-L2-s128* retains a high average score of 89.9, far surpassing other methods. This robustness makes *SparseCache* a more reliable solution for practical systems.

*SparseCache* is the only method that provides functional performance at compression ratios where other techniques fail completely. This is a critical advantage for enabling the "precomputed RAG" paradigm on massive corpora where storage is the primary constraint. As shown in the results for the *Qwen2.5-14B* model, at a 17.7× compression ratio, *SparseCache-L4-s128* still achieves a remarkable 82.2 average score. At this level of compression, all other baseline methods have effectively failed. This unique capability positions *SparseCache* as an enabling technology for next-generation, low-latency RAG systems.

## 5.3 ANALYSIS OF SPARSECACHE HYPERPARAMETERS

The detailed results in Table 2 highlight the controllable nature of the *SparseCache* framework. The primary hyperparameter for controlling the compression-accuracy trade-off is the sparsity level $s$. Across all $L$ configurations, decreasing $s$ (using fewer dictionary atoms) consistently leads to a higher compression ratio at the cost of some performance. For example, in the *SparseCache-L4* series on *Qwen2.5-7B*:

- 's128' ( 8.8× comp.) yields 84.7% accuracy.
- 's64' ( 17.7× comp.) yields 62.0% accuracy.
- 's32' ( 35.3× comp.) yields 32.7% accuracy.

This predictable behavior allows practitioners to tune *SparseCache* precisely to their specific hardware limitations and performance requirements, selecting a configuration that offers the best possible accuracy for a given memory budget.

Furthermore, the 'L' parameter, which dictates the degree of layer-wise dictionary sharing, also plays a crucial role in balancing structural compression with performance. A higher $L$ (e.g., $L4$) leads to fewer global dictionaries, thus achieving higher overall compression by leveraging more pervasive redundancies across multiple layers. While this might introduce a slight performance trade-off compared to layer-specific dictionaries ($L1$) at very low compression ratios, it significantly enhances the scalability and storage efficiency for large-scale "precomputed RAG" systems, especially in high-compression regimes where its structural efficiency becomes dominant.

## 5.4 SYSTEM PERFORMANCE AND MEMORY FOOTPRINT

We analyze the impact of precomputed KV Cache and compression strategies on Time-To-First-Token (TTFT) and memory footprint under various system configurations. Although OMP-based

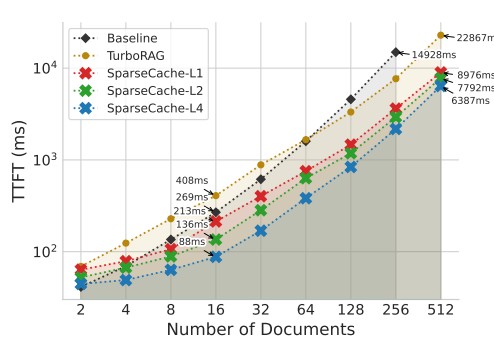
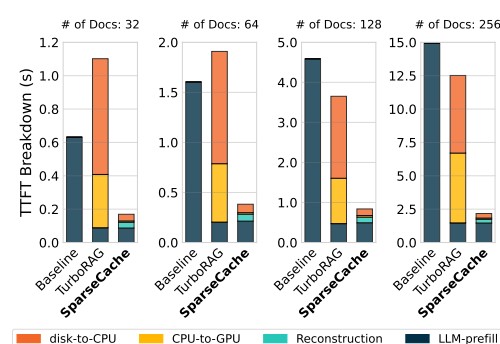

(a) TTFT vs. Number of Documents. Comparison of different KV Cache prefill strategies on the *Qwen2.5-7B-Instruct-1M* model. The *Baseline* method fails with an Out-of-Memory (OOM) error at 512 documents.

(b) TTFT breakdown into its constituent processing stages-disk-to-CPU, CPU-to-GPU, Reconstruction, and LLM-prefill—comparing *Baseline*, *TurboRAG*, and *SparseCache-L4-s64* for 32, 64, 128, and 256 retrieved documents.

Figure 4: System performance analysis. (a) Time-To-First-Token (TTFT) comparison across different methods. (b) A breakdown of TTFT latency into its primary stages.

compression is executed entirely offline, it substantially contributes to reducing TTFT in RAG systems by enabling faster loading of precomputed KV Caches during the prefill phase. The computational complexity of the OMP-based compression per token is $\mathcal{O}(s \cdot m \cdot n)$, where $s$ is the sparsity level, $m$ is the dimensionality of the KV vector, and $n$ is the dictionary size. For a batch of $B$ token KV Cache vectors, the total offline cost becomes $\mathcal{O}(B \cdot s \cdot m \cdot n)$, and the token-wise independence of OMP makes the compression process efficiently parallelizable on modern GPUs.

Figure 4a reports TTFT measurements for loading KV Caches from disk on a single NVIDIA H100 GPU using the *Qwen2.5-7B-Instruct-1M* model, with each document comprising 512 tokens. In this experiment, *Baseline* represents direct KV computation during the prefill phase without precomputed KV Cache, *TurboRAG* denotes the use of precomputed KV Caches without compression, whereas *SparseCache-L1/L2/L4* represents our proposed method that combines precomputation with compression of KV Caches. Although absolute latency values may vary across hardware, consistent trends were observed.

While *Baseline* performs well when the number of retrieved documents is very small (e.g., 2 documents), its TTFT grows rapidly as the number of documents increases because all KV Caches must be computed online, and it eventually encounters an Out-of-Memory (OOM) error beyond 256 documents (131072 tokens), highlighting its memory scalability limitations compared to *TurboRAG* and *SparseCache*.

*TurboRAG* benefits from using precomputed KV Caches, but the advantage is offset by the overhead of loading large KV Cache files when the number of documents is small. For fewer than 64 documents, *TurboRAG*'s TTFT can be comparable to or even worse than *Baseline*. However, as the number of documents grows, the reuse of precomputed KV Caches allows *TurboRAG* to achieve lower TTFT than *Baseline*, corresponding to about a $1.3\times$ speedup at 128 documents.

On the other hand, *SparseCache* provides the most stable and substantial performance gains. Once the number of documents reaches 8, all *SparseCache-L1/L2/L4* variants outperform both *Baseline* and *TurboRAG*. As the number of documents increases, *SparseCache* achieves $5.8\times$ lower TTFT than *TurboRAG* and $6.9\times$ lower than *Baseline* at 256 documents. At 512 documents, where *Baseline* runs out of memory, *SparseCache* still maintains $3.5\times$ lower TTFT than *TurboRAG*.

Following the overall TTFT trends shown in Figure 4a, Figure 4b further breaks down the TTFT latency into its constituent stages to show how the use of precomputed KV Caches and compression reshapes the distribution of system resource usage. While *TurboRAG* reduces the latency of `LLM-prefill`, the breakdown shows that the saved computation is almost completely replaced by KV Cache loading overhead (`disk-to-CPU` and `CPU-to-GPU`). In contrast, *SparseCache* compresses precomputed KV Caches to greatly reduce their data size, which in turn leads to up to a

96.1% reduction in KV Cache loading time compared to *TurboRAG*. Although *SparseCache* adds a `Reconstruction` step for decompressing the sparse codes, the overhead of `Reconstruction` has only a minor impact on TTFT, as it accounts for 13% of the total latency at 256 documents. Consequently, *SparseCache* makes "precomputed RAG" practical and efficient for large-scale inference systems.

We also analyze the memory footprint of the KV Caches during inference. *TurboRAG*, which stores uncompressed KV Caches, results in rapidly growing memory usage, highlighting its scalability limitations in long-context scenarios. In contrast, *SparseCache* significantly reduces memory consumption by compressing KV Caches into sparse representations. This enables scalable, memory-efficient inference, making it practical for deployment in resource-constrained environments. For a detailed breakdown and analysis of the memory footprint, please refer to Appendix A.8.

It is important to note that our reported compression ratios include the dictionary storage overhead. For *Qwen2.5-7B*, the total dictionary size (both Key and Value) is only 16.8 MB. When processing 512 documents with our best compression ($17.7\times$), the compressed KV Cache requires approximately 790 MB (down from 14.0 GB uncompressed). The dictionary overhead represents just 2.1% of the total compressed size (dictionary + compressed cache). Even in this worst-case scenario, the effective compression ratio remains $16.9\times$ when accounting for dictionary storage. For larger corpora, this overhead becomes negligible as the dictionary is amortized across all documents, confirming that our reported compression ratios are both realistic and conservative.

## 6  CONCLUSION AND FUTURE WORK

This paper introduced *SparseCache*, a novel KV Cache compression method that leverages K-SVD dictionaries learned in an end-to-end manner, specifically tailored for LLMs. Our end-to-end learned global dictionary approach proves highly effective by training dictionaries using deep-learning paradigms on large text datasets and optimizing for reconstruction fidelity. This enables *SparseCache* to achieve state-of-the-art compression ratios while robustly preserving LLM accuracy, making "precomputed RAG" paradigms practical for the first time.

Several avenues for future research exist:

- Exploring more advanced or efficient sparse coding and dictionary update algorithms within the *SparseCache* framework.

- Developing theoretical analyses of the properties of end-to-end learned KV Cache dictionaries.

- Combining *SparseCache* with orthogonal compression techniques (e.g., further quantizing the learned sparse codes or the dictionary itself).

- Exploring hardware co-design for *SparseCache*-like compression.

- Studying the generalization capabilities of *SparseCache*-learned dictionaries across different tasks, domains, and LLM architectures to understand the trade-off between universal and specialized dictionaries.

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

# A   APPENDIX

## A.1   DICTIONARY LEARNING HYPERPARAMETERS

The dictionaries used in our experiments were trained using the hyperparameters detailed in Table 1. The training was performed on a machine equipped with 8 NVIDIA H100 GPUs.

Table 1: Dictionary Learning Hyperparameters

| Hyperparameter | Value |
|---|---|
| model | Llama3.1-8B-Instruct |
| | Qwen2.5-7B-Instruct-1M |
| | Qwen2.5-14-Instruct-1M |
| dictionary size | 8192 |
| sparsity | [32, 48, 64, 128, 256] |
| L (# consecutive layers) | [1, 2, 4] |
| # epochs | 3 |
| batch size | 1024 |
| learning rate | 0.0005 |
| use-norm | True |

## A.2   DETAILED FORMULATION OF THE K-SVD ALGORITHM

This section provides a more detailed description of the classic K-SVD algorithm (Aharon et al., 2006), which serves as a foundation for the SparseCache framework. As mentioned in the main text, K-SVD is an iterative method that alternates between a sparse coding stage and a dictionary update stage.

### A.2.1   SPARSE CODING STAGE USING OMP

Given a fixed dictionary $\mathbf{D}$, this stage aims to find the optimal sparse coefficient matrix $\mathbf{X}$ for the input signals $\mathbf{Y}$. This is done by solving the following problem for each signal $\mathbf{y}_i$:

$$\min_{\mathbf{x}_i} \|\mathbf{y}_i - \mathbf{D}\mathbf{x}_i\|_2^2 \quad \text{s.t.} \quad \|\mathbf{x}_i\|_0 \leq s \tag{6}$$

K-SVD typically employs Orthogonal Matching Pursuit (OMP) for this task. OMP is a greedy algorithm that, for a given signal $\mathbf{y}$ and dictionary $\mathbf{D}$, iteratively selects the dictionary atom $\mathbf{d}_l$ that is most correlated with the current residual. This selected atom is added to a support set $\mathcal{S}$. The signal is then orthogonally projected onto the subspace spanned by the atoms in $\mathcal{S}$ to find new coefficients, and the residual is updated. This process repeats until $s$ atoms have been selected.

### A.2.2   DICTIONARY UPDATE STAGE

With the sparse coefficient matrix $\mathbf{X}$ fixed, the dictionary $\mathbf{D}$ is updated. K-SVD updates one atom (column $\mathbf{d}_k$) at a time. For each atom $\mathbf{d}_k$, it first identifies the set of signals $\omega_k$ that use this atom in their sparse representation (i.e., the $k$-th coefficient in $\mathbf{x}_i$ is nonzero):

$$\omega_k = \{i \mid \mathbf{x}_k^T(i) \neq 0\} \tag{7}$$

where $\mathbf{x}_k^T$ is the $k$-th row of $\mathbf{X}$. The update aims to minimize the reconstruction error for these specific signals by solving $\left\|\mathbf{E}_k - \mathbf{d}_k\mathbf{x}_k^T\right\|_F^2$. Here, $\mathbf{E}_k$ is the error matrix calculated by removing the contribution of all other atoms for the signals in $\omega_k$:

$$\mathbf{E}_k = \mathbf{Y}_{\omega_k} - \sum_{j \neq k} \mathbf{d}_j\mathbf{x}_j^T|_{\omega_k} \tag{8}$$

This minimization subproblem is solved by applying Singular Value Decomposition (SVD) to a restricted error matrix derived from $\mathbf{E}_k$. The atom $\mathbf{d}_k$ is updated to be the first left singular vector,

---

**Algorithm 1:** SparseCache: Global Dictionary Learning via Alternating Optimization

---

**Input:** KV Cache dataset for training $(\mathbf{Y}_K^{\text{train}}, \mathbf{Y}_V^{\text{train}})$; sparsity $s$; dictionary size $n$; learning rate $\eta$

**Output:** $\mathbf{D}_K^*, \mathbf{D}_V^*$ - Learned global SparseCache dictionaries

1 Initialize $\mathbf{D}_K, \mathbf{D}_V$ (e.g., Kaiming uniform, normalize columns);

2 **while** *until convergence* **do**

3      Sample $(\mathbf{Y}_{K,\text{batch}}, \mathbf{Y}_{V,\text{batch}})$ from $(\mathbf{Y}_K^{\text{train}}, \mathbf{Y}_V^{\text{train}})$ ;

4      $\mathbf{D}_K \leftarrow \text{UpdateDictWithGD}(\mathbf{Y}_{K,\text{batch}}, \mathbf{D}_K, s, \eta)$;

5      $\mathbf{D}_V \leftarrow \text{UpdateDictWithGD}(\mathbf{Y}_{V,\text{batch}}, \mathbf{D}_V, s, \eta)$;

6 $(\mathbf{D}_K^*, \mathbf{D}_V^*) \leftarrow (\mathbf{D}_K, \mathbf{D}_V)$;

7 **return** $\mathbf{D}_K^*, \mathbf{D}_V^*$

8 **Procedure** *UpdateDictWithGD($\mathbf{Y}_{batch}, \mathbf{D}, s, \eta$)* **is**

9      // Sparse Coding Stage

     $\mathbf{X}_{\text{batch}} \leftarrow \text{OMP}(\mathbf{Y}_{\text{batch}}, \mathbf{D}, s)$;

     // Dictionary Update Stage (Gradient-based)

10      $\mathbf{G} \leftarrow \nabla_{\mathbf{D}} \|\mathbf{Y}_{\text{batch}} - \mathbf{D}\mathbf{X}_{\text{batch}}\|_F^2$;

11      $\mathbf{D} \leftarrow \mathbf{D} - \eta\mathbf{G}$;

12      $\mathbf{d}_j \leftarrow \mathbf{d}_j / \|\mathbf{d}_j\|_2 \quad$ for $j = 1, \dots, n$;

13      **return** $\mathbf{D}$;

---

**Algorithm 2:** SparseCache: Offline Corpus Compression

---

**Input:** Dicts $(\mathbf{D}_K^*, \mathbf{D}_V^*)$; Corpus $C_{\text{RAG}}$; Model $\mathcal{M}$; Sparsity $s$

**Output:** $\mathcal{C}_{KV}$ - Collection of sparse codes

1 $\mathcal{C}_{KV} \leftarrow \emptyset$ // Initialize storage

2 $\mathbf{G}_K \leftarrow (\mathbf{D}_K^*)^T \mathbf{D}_K^*$ // Pre-compute for OMP

3 $\mathbf{G}_V \leftarrow (\mathbf{D}_V^*)^T \mathbf{D}_V^*$

4 **foreach** *doc* $\in C_{RAG}$ **do**

5      chunks $\leftarrow \text{SplitChunks}(doc)$;

6      **foreach** $(chunk, id) \in enumerate(chunks)$ **do**

7          // Generate KV vectors

         $(\mathbf{Y}_K, \mathbf{Y}_V) \leftarrow \text{PrepKVCache}(\mathcal{M}, \text{chunk})$;

         // Compress K and V vectors

8          $(\mathbf{ids}_K, \mathbf{scale}_K) \leftarrow \text{OMP}(\mathbf{Y}_K, \mathbf{D}_K^*, \mathbf{G}_K, s)$;

9          $(\mathbf{ids}_V, \mathbf{scale}_V) \leftarrow \text{OMP}(\mathbf{Y}_V, \mathbf{D}_V^*, \mathbf{G}_V, s)$;

         // Store sparse codes

10          $\mathcal{C}_{doc,K} \leftarrow \mathcal{C}_{doc,K} \cup \{(\mathbf{ids}_K, \mathbf{scale}_K)\}$;

11          $\mathcal{C}_{doc,V} \leftarrow \mathcal{C}_{doc,V} \cup \{(\mathbf{ids}_V, \mathbf{scale}_V)\}$;

12          $\mathcal{C}_{KV} \leftarrow \mathcal{C}_{doc,K}, \mathcal{C}_{doc,V}$

13 **return** $\mathcal{C}_{KV}$

14 **Function** *PrepKVCache($\mathcal{M}$, chunk)*:

15      raw_kv $\leftarrow \mathcal{M}.\text{get\_kv\_cache}(\text{chunk})$;

16      $(\mathbf{Y}_K, \mathbf{Y}_V) \leftarrow \text{ReshapeAndMerge}(\text{raw\_kv})$;

17      **return** $(\mathbf{Y}_K, \mathbf{Y}_V)$

---

and the corresponding coefficients in $\mathbf{x}_k^T$ are updated using the first right singular vector and the largest singular value. We note that while SparseCache is inspired by this structure, its dictionary update step is implemented via mini-batch gradient descent rather than the atom-wise SVD update described here.

## A.3 SparseCache Algorithms

This section contains the pseudocode for the core components of the SparseCache framework, which were referenced in the Methods section.

---

**Algorithm 3:** SparseCache: KV Cache Decompression for RAG Query Time

---

**Input:** Dictionaries $\mathbf{D}_K^*, \mathbf{D}_V^*$; Set of retrieved codes $\mathcal{C}_{\text{retrieved}} \subseteq \mathcal{C}_{KV}$
**Output:** $\mathbf{KV}_{recon}$ - Reconstructed dense KV Cache matrices

1   $\mathbf{KV}_{recon} \leftarrow \emptyset$
2   **foreach** $(\mathcal{C}_{doc,K}, \mathcal{C}_{doc,V}) \in \mathcal{C}_{\text{retrieved}}$ **do**
3      Initialize $\hat{\mathbf{Y}}_{doc,K}, \hat{\mathbf{Y}}_{doc,V}$
4      **foreach** $(\mathbf{ids}, \mathbf{scale}) \in \mathcal{C}_{doc,K}$ **do**
5          $\hat{\mathbf{y}}_K \leftarrow \sum_{i=1}^{s} \mathbf{D}_{K,\mathbf{ids}(i)}^* \cdot \mathbf{scale}(i)$
6          Append $\hat{\mathbf{y}}_K$ to $\hat{\mathbf{Y}}_{doc,K}$
7      **foreach** $(\mathbf{ids}, \mathbf{scale}) \in \mathcal{C}_{doc,V}$ **do**
8          $\hat{\mathbf{y}}_V \leftarrow \sum_{i=1}^{s} \mathbf{D}_{V,\mathbf{ids}(i)}^* \cdot \mathbf{scale}(i)$
9          Append $\hat{\mathbf{y}}_V$ to $\hat{\mathbf{Y}}_{doc,V}$
10      Append $(\hat{\mathbf{Y}}_{doc,K}, \hat{\mathbf{Y}}_{doc,V})$ to $\mathbf{KV}_{recon}$

---

## A.4 Detailed Performance Comparison on RULER Benchmark

This section provides a comprehensive breakdown of the performance of various KV Cache compression methods on the RULER benchmark, with detailed results presented in Table 2. The evaluation spans three different large language models: **Llama3.1-8B-Instruct**, **Qwen2.5-7B-Instruct-1M**, and **Qwen2.5-14B-Instruct-1M**. The table compares our proposed *SparseCache* method against several baselines, including *Minicache*, *Single SVD*, and *xKV-4*, across all sub-tasks of the benchmark as well as the average score.

The results highlight several key trends. At lower compression ratios (approx. 2-4×, highlighted in light green), most advanced methods, including *SparseCache*, successfully maintain performance close to the uncompressed baseline, demonstrating high fidelity. As the compression ratio increases to a medium range (approx. 8×, highlighted in light pink), a performance drop becomes more apparent across all methods. However, *SparseCache* variants generally exhibit more graceful degradation compared to methods like *Single SVD*. The most significant advantage of our method is observed at very high compression ratios (17.7× and above, highlighted in light blue). In this regime, where other methods either fail or suffer a catastrophic loss in accuracy, *SparseCache* (e.g., *SparseCache-L4-s128*) continues to deliver strong, usable performance. This demonstrates its superior capability in balancing high compression with model accuracy, making it a robust solution for memory-constrained environments.

## A.5 Additional Implementation Details

This section provides additional details regarding hyperparameter considerations and the implications of layer-wise dictionary specialization for the SparseCache framework.

### A.5.1 Hyperparameter Considerations

- **Initialization of Dictionary** $D$: The performance of dictionary learning can be sensitive to initialization. Options include random selection from training samples, output of K-means, or a pre-trained universal dictionary. For SparseCache, dictionaries were initialized using Kaiming uniform initialization.

- **Number of Training Iterations**: Too few iterations may result in a suboptimal dictionary, while too many increase training time with diminishing returns. This needs to be tuned based on the convergence of the dictionary and the end-to-end objective. Our training typically converged within 10,000 steps.

Table 2: Detailed performance comparison of various KV Cache compression methods on the RULER benchmark across different models. Scores are reported for individual sub-tasks as well as the average. *Comp.* denotes the compression ratio. We highlight strong performing methods at lower compression (light green), higher compression (light pink), and our method at a very high compression ratio (light blue) for easier comparison.

| Method | Comp. | N-S1 | N-S2 | N-MK1 | N-MK2 | N-MQ | N-MV | QA-1 | QA-2 | VT | FWE | Avg. |
|---|---|---|---|---|---|---|---|---|---|---|---|---|
| **Llama3.1-8B-Instruct** | | | | | | | | | | | | |
| Baseline | 1.0 | 100.0 | 100.0 | 99.0 | 97.9 | 99.0 | 98.4 | 83.3 | 60.4 | 97.3 | 84.7 | 92.0 |
| Minicache | 1.2 | 100.0 | 100.0 | 97.9 | 90.6 | 87.0 | 81.0 | 78.1 | 47.9 | 84.6 | 84.0 | 85.1 |
| Minicache | 1.3 | 87.5 | 64.6 | 39.6 | 10.4 | 13.3 | 20.1 | 60.4 | 35.4 | 49.0 | 58.0 | 43.8 |
| Single SVD | 2.5 | 100.0 | 100.0 | 100.0 | 97.9 | 97.9 | 96.1 | 80.2 | 58.3 | 96.9 | 79.5 | 90.7 |
| Single SVD | 8.4 | 29.2 | 26.0 | 32.3 | 96.9 | 8.6 | 17.2 | 44.8 | 36.5 | 2.7 | 59.0 | 35.3 |
| xKV-4 | 2.4 | 100.0 | 100.0 | 100.0 | 97.9 | 98.4 | 97.1 | 84.4 | 60.4 | 96.2 | 81.2 | 91.6 |
| xKV-4 | 8.0 | 100.0 | 96.9 | 97.9 | 97.9 | 95.3 | 93.5 | 76.0 | 54.2 | 87.7 | 78.8 | 87.8 |
| SparseCache-L1-s128 | 4.4 | 100.0 | 100.0 | 97.9 | 95.8 | 99.5 | 99.2 | 80.2 | 57.3 | 92.9 | 85.4 | 90.8 |
| SparseCache-L1-s64 | 8.8 | 93.8 | 74.0 | 76.0 | 64.6 | 69.3 | 73.7 | 69.8 | 51.0 | 81.0 | 75.3 | 72.9 |
| SparseCache-L2-s256 | 4.4 | 100.0 | 100.0 | 99.0 | 94.8 | 99.7 | 99.0 | 82.3 | 56.3 | 92.5 | 88.5 | 91.2 |
| SparseCache-L2-s128 | 8.8 | 97.9 | 96.9 | 91.7 | 89.6 | 90.9 | 94.5 | 79.2 | 50.0 | 90.6 | 85.4 | 86.7 |
| SparseCache-L2-s64 | 17.7 | 94.8 | 62.5 | 41.7 | 9.4 | 34.9 | 34.9 | 53.1 | 50.0 | 52.5 | 60.8 | 49.5 |
| SparseCache-L4-s256 | 8.8 | 100.0 | 96.9 | 90.6 | 87.5 | 95.3 | 94.5 | 77.1 | 50.0 | 92.1 | 82.3 | 86.6 |
| SparseCache-L4-s128 | 17.7 | 95.8 | 69.8 | 53.1 | 20.8 | 47.4 | 51.0 | 58.3 | 47.9 | 45.8 | 68.8 | 55.9 |
| SparseCache-L4-s64 | 35.3 | 61.5 | 13.5 | 2.1 | 1.0 | 0.5 | 2.6 | 45.8 | 39.6 | 17.9 | 39.2 | 22.4 |
| **Qwen2.5-7B-Instruct-1M** | | | | | | | | | | | | |
| Baseline | 1.0 | 100.0 | 100.0 | 100.0 | 100.0 | 100.0 | 95.6 | 83.3 | 59.4 | 90.8 | 86.5 | 91.7 |
| Minicache | 1.2 | 78.1 | 34.4 | 32.3 | 3.1 | 27.3 | 42.7 | 26.0 | 24.0 | 8.5 | 8.0 | 28.4 |
| Minicache | 1.3 | 25.0 | 0.0 | 0.0 | 0.0 | 0.0 | 0.0 | 13.5 | 13.5 | 0.8 | 4.5 | 5.7 |
| Single SVD | 2.3 | 100.0 | 100.0 | 99.0 | 99.0 | 99.7 | 92.7 | 75.0 | 58.3 | 80.8 | 75.7 | 88.0 |
| Single SVD | 6.4 | 24.0 | 7.3 | 6.2 | 97.9 | 4.4 | 3.4 | 36.5 | 35.4 | 6.9 | 26.4 | 24.8 |
| xKV-4 | 2.2 | 100.0 | 100.0 | 100.0 | 99.0 | 100.0 | 90.9 | 83.3 | 60.4 | 85.0 | 83.0 | 90.2 |
| xKV-4 | 6.2 | 100.0 | 85.4 | 91.7 | 99.0 | 84.1 | 76.8 | 62.5 | 50.0 | 66.2 | 64.9 | 78.1 |
| SparseCache-L1-s64 | 4.4 | 100.0 | 100.0 | 97.9 | 100.0 | 99.5 | 94.0 | 79.2 | 57.3 | 89.8 | 81.6 | 89.9 |
| SparseCache-L1-s48 | 5.8 | 100.0 | 100.0 | 97.9 | 97.9 | 98.7 | 94.3 | 79.2 | 54.2 | 66.7 | 81.9 | 87.1 |
| SparseCache-L1-s32 | 8.8 | 99.0 | 89.6 | 92.7 | 92.7 | 91.9 | 83.3 | 64.6 | 51.0 | 41.7 | 75.0 | 78.2 |
| SparseCache-L2-s64 | 8.8 | 100.0 | 95.8 | 95.8 | 92.7 | 97.4 | 85.9 | 65.6 | 50.0 | 65.4 | 82.6 | 83.1 |
| SparseCache-L2-s48 | 11.7 | 99.0 | 89.6 | 94.8 | 75.0 | 94.8 | 82.6 | 50.0 | 49.0 | 57.1 | 76.4 | 76.8 |
| SparseCache-L2-s32 | 17.7 | 95.8 | 69.8 | 66.7 | 36.5 | 64.8 | 61.7 | 40.6 | 45.8 | 34.6 | 66.7 | 58.3 |
| SparseCache-L4-s128 | 8.8 | 100.0 | 99.0 | 95.8 | 83.3 | 97.9 | 87.2 | 62.5 | 55.2 | 84.6 | 81.6 | 84.7 |
| SparseCache-L4-s64 | 17.7 | 100.0 | 83.3 | 66.7 | 15.6 | 77.6 | 83.6 | 41.7 | 38.5 | 55.6 | 56.9 | 62.0 |
| SparseCache-L4-s48 | 23.4 | 92.7 | 86.5 | 68.8 | 17.7 | 71.1 | 72.7 | 40.6 | 40.6 | 26.3 | 49.3 | 56.6 |
| SparseCache-L4-s32 | 35.3 | 79.2 | 42.7 | 21.9 | 3.1 | 24.2 | 58.6 | 30.2 | 29.2 | 16.3 | 21.5 | 32.7 |
| **Qwen2.5-14B-Instruct-1M** | | | | | | | | | | | | |
| Baseline | 1.0 | 100.0 | 100.0 | 100.0 | 99.0 | 100.0 | 99.2 | 80.2 | 66.7 | 99.6 | 91.3 | 93.6 |
| Minicache | 1.2 | 100.0 | 100.0 | 100.0 | 99.0 | 98.2 | 99.0 | 72.9 | 61.5 | 90.8 | 84.4 | 90.6 |
| Minicache | 1.3 | 0.0 | 1.0 | 0.0 | 0.0 | 0.0 | 0.0 | 17.7 | 27.1 | 0.2 | 8.3 | 5.4 |
| Single SVD | 2.5 | 100.0 | 100.0 | 100.0 | 99.0 | 100.0 | 96.6 | 78.1 | 62.5 | 98.5 | 87.2 | 92.2 |
| Single SVD | 6.4 | 24.0 | 7.3 | 6.2 | 97.9 | 4.4 | 3.4 | 36.5 | 35.4 | 6.9 | 26.4 | 24.8 |
| xKV-4 | 2.4 | 100.0 | 100.0 | 100.0 | 99.0 | 100.0 | 95.3 | 83.3 | 69.8 | 99.4 | 88.5 | 93.5 |
| xKV-4 | 8.0 | 100.0 | 96.9 | 99.0 | 97.9 | 97.1 | 88.5 | 63.5 | 58.3 | 86.0 | 86.5 | 87.4 |
| SparseCache-L1-s128 | 4.4 | 100.0 | 100.0 | 100.0 | 99.0 | 100.0 | 96.9 | 78.1 | 63.5 | 99.2 | 89.9 | 92.7 |
| SparseCache-L1-s64 | 8.8 | 95.8 | 88.5 | 96.9 | 96.9 | 97.7 | 90.1 | 64.6 | 60.4 | 92.1 | 81.6 | 86.5 |
| SparseCache-L2-s256 | 4.4 | 100.0 | 100.0 | 100.0 | 97.9 | 100.0 | 97.4 | 78.1 | 63.5 | 99.4 | 90.6 | 92.7 |
| SparseCache-L2-s128 | 8.8 | 100.0 | 100.0 | 99.0 | 95.8 | 99.5 | 94.8 | 66.7 | 61.5 | 98.1 | 83.7 | 89.9 |
| SparseCache-L2-s64 | 17.7 | 92.7 | 59.4 | 82.3 | 62.5 | 4.4 | 3.9 | 41.7 | 51.0 | 80.2 | 0.0 | 47.8 |
| SparseCache-L4-s256 | 8.8 | 100.0 | 100.0 | 100.0 | 96.9 | 99.0 | 95.6 | 75.0 | 59.4 | 100.0 | 87.2 | 91.3 |
| SparseCache-L4-s128 | 17.7 | 100.0 | 94.8 | 99.0 | 82.3 | 93.5 | 83.6 | 58.3 | 53.1 | 89.4 | 67.7 | 82.2 |
| SparseCache-L4-s64 | 35.3 | 71.9 | 52.1 | 60.4 | 5.2 | 32.0 | 26.0 | 36.5 | 43.8 | 32.1 | 0.0 | 36.0 |

- **Calibration Data for Dictionary Learning**: The choice and size of the dataset used to generate KV vectors for dictionary training are crucial. As detailed in Section 4.0.1, we used a subset of the Wikitext dataset, which was selected to be representative of general text distributions relevant to LLM applications.

### A.5.2 POTENTIAL FOR LAYER-WISE DICTIONARY SPECIALIZATION

It is known that KV Cache statistics can vary significantly across LLM layers (Liu et al., 2024a). Although SparseCache employs global dictionaries, the end-to-end learning process can naturally adapt to these variations. Our results, particularly the performance of 'SparseCache-L4' which shares a single dictionary across every four layers, suggest that a global dictionary can effectively capture universal patterns across different layers without severe performance degradation. This indicates that while layer-specific nuances exist, there are also strong underlying commonalities in KV Cache structures that a well-trained global dictionary can exploit for compression. Future work could explore mechanisms to implicitly or explicitly encourage specialization within a global dictionary, or dynamically decide on layer grouping based on learned characteristics.

### A.6 INDEX COMPRESSION FOR ENHANCED STORAGE EFFICIENCY

To further improve the compression ratio, we implement an additional optimization for storing sparse code indices. Since our dictionary size is fixed at 8192 atoms, each index requires only 13 bits (as $2^{13} = 8192$). However, standard integer types use 16 bits (int16), wasting 3 bits per index. We exploit this redundancy through a bit-packing algorithm that compresses indices from 16 bits to 13 bits.

The core idea is to treat the indices as a continuous stream of 13-bit values and repack them into bytes without gaps. We process indices in groups of 8 because the least common multiple of 8 (bits per byte) and 13 (bits per index) is 104 bits. This means 8 indices can be perfectly packed from 128 bits ($8 \times 16$) into 104 bits ($13 \times 8$), achieving exactly 13 bytes with no wasted space.

The challenge lies in the fact that 13-bit values don't align with byte boundaries. When packing, each index's bits must be distributed across one or two bytes. For example, the first index uses all 8 bits of the first byte plus 5 bits of the second byte, while the second index uses the remaining 3 bits of the second byte plus all of the third byte and 2 bits of the fourth byte. This pattern continues throughout the packed representation.

Algorithm 4 shows the packing process. For each group of 8 indices, we extract their 13-bit representations and redistribute the bits across 13 bytes using bitwise operations. The unpacking process (not shown) reverses these operations to recover the original indices.

---

**Algorithm 4:** Index Compression: 16-bit to 13-bit Packing

**Input:** Sparse code indices $\mathbf{ids} \in \{0, ..., 8191\}^n$ (stored as int16 array)
**Output:** Packed indices **packed** (stored as uint8 array)
1   Initialize empty byte array **packed**;
2   **foreach** *group of 8 consecutive indices in* **ids do**
3      Extract indices $[idx_0, idx_1, ..., idx_7]$;
4      Create bit stream by concatenating 13-bit values:;
5      $bitstream \leftarrow idx_0[12:0] \,\|\, idx_1[12:0] \,\|\, ... \,\|\, idx_7[12:0]$;
6      Distribute 104 bits into 13 bytes:;
7      **for** $i = 0$ *to* 12 **do**
8         $start\_bit \leftarrow i \times 8$;
9         $bytes[i] \leftarrow$ Extract bits $[start\_bit : start\_bit + 8]$ from $bitstream$;
10      Append $bytes[0..12]$ to **packed**;
11   **return packed**

---

This bit-packing achieves a 1.23× compression ratio (16/13) for the index component alone. When combined with the quantization of scale values and the sparsity inherent in our representation, this

index compression contributes significantly to our overall compression ratios of up to 17.7×. The technique is particularly effective because these bit-packing and unpacking operations are highly parallelizable on modern GPUs, adding negligible overhead to the reconstruction process while providing meaningful storage savings.

## A.7 DETAILED TTFT RESULTS ACROSS ALL SETTINGS

Table 3 presents the complete Time-To-First-Token (TTFT) measurements for all evaluated settings under varying numbers of documents, each with a context length of 512 tokens. This table complements Figure 4a in the main paper by including configurations that could not be shown due to space limitations.

As the number of documents increases, TTFT generally grows across all methods, but the rate of increase varies significantly. SparseCache consistently achieves lower TTFT than both the baseline without prefill and the uncompressed KV Cache. Among SparseCache variants, higher compression levels (e.g., L4) combined with smaller sparsity (e.g., s32) tend to produce the fastest response times. For example, 'SparseCache-L4-s32' records a TTFT of $6,023$ms when prefilling $262,144$ tokens (512 documents), compared to $22,866$ms for the uncompressed KV Cache. This illustrates not only the computational efficiency of SparseCache but also its scalability with respect to context length. The performance gap widens at scale, emphasizing the advantage of compression-based KV caching in memory- or latency-sensitive deployments.

Table 3: TTFT (Time-To-First-Token) across all experimental settings in milliseconds, including baseline (no prefill), uncompressed KV Cache, and SparseCache with various compression levels. Each entry reports latency measured when pre-filling a 512-token document on Qwen2.5-7B-Instruct-1M. The '-' indicates an Out-of-Memory (OOM) error.

| # docs | 2 | 4 | 8 | 16 | 32 | 64 | 128 | 256 | 512 |
|---|---|---|---|---|---|---|---|---|---|
| Baseline | 41 | 69 | 135 | 268 | 615 | 1598 | 4576 | 14928 | - |
| TurboRAG | 69 | 124 | 228 | 407 | 883 | 1660 | 3326 | 7662 | 22866 |
| SparseCache-L1-s64 | 58 | 74 | 98 | 198 | 370 | 684 | 1261 | 3315 | 8352 |
| SparseCache-L1-s48 | 50 | 73 | 93 | 168 | 342 | 590 | 1247 | 2973 | 7819 |
| SparseCache-L1-s32 | 48 | 62 | 82 | 150 | 284 | 543 | 1111 | 3544 | 7545 |
| SparseCache-L2-s64 | 49 | 63 | 81 | 120 | 250 | 552 | 1032 | 2620 | 7125 |
| SparseCache-L2-s48 | 45 | 61 | 81 | 109 | 247 | 531 | 973 | 2478 | 6956 |
| SparseCache-L2-s32 | 44 | 57 | 71 | 97 | 213 | 449 | 874 | 2610 | 6551 |
| SparseCache-L4-s128 | 44 | 53 | 74 | 116 | 213 | 454 | 1041 | 2468 | 7117 |
| SparseCache-L4-s64 | 44 | 49 | 63 | 87 | 169 | 382 | 838 | 2173 | 6386 |
| SparseCache-L4-s48 | 41 | 46 | 60 | 89 | 163 | 350 | 300 | 2068 | 6154 |
| SparseCache-L4-s32 | 39 | 44 | 60 | 82 | 145 | 341 | 763 | 2004 | 6023 |

## A.8 DETAILED MEMORY FOOTPRINT ANALYSIS

Figure 5 shows the memory footprint of the KV Caches during inference as the number of retrieved documents increases, comparing *TurboRAG* with different *SparseCache* variants.

As shown in Figure 5, *TurboRAG*—where uncompressed KV Caches are stored for each layer—results in rapidly growing memory usage, reaching up to 14.0 GB with 512 documents. This highlights the scalability limitations of uncompressed caching in long-context scenarios. In contrast, *SparseCache* significantly reduces memory consumption by compressing KV Caches into sparse representations using globally learned dictionaries. Among all configurations, *SparseCache-L4* achieves the lowest memory footprint—just 0.79 GB at 512 documents—representing 17.7× reduction compared to the *TurboRAG*. These results demonstrate that *SparseCache* enables scalable, memory-efficient inference by structurally alleviating system-level memory demands, making it practical for deployment in resource-constrained environments.

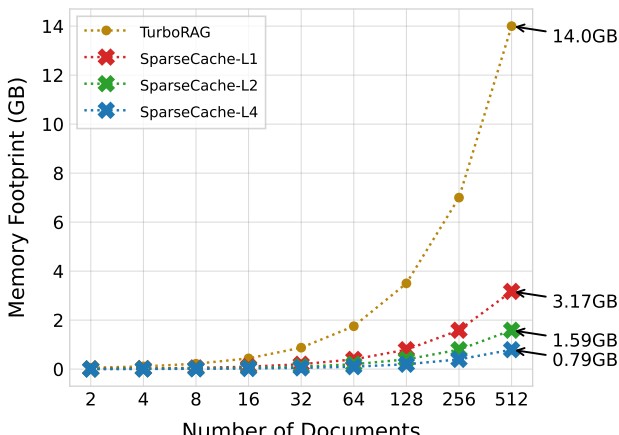

Figure 5: Memory footprint of the KV Cache under *TurboRAG* and different *SparseCache* variants as the number of documents increases. Evaluation is conducted on *Qwen2.5-7B-Instruct-1M* with sparsity = 64.

## A.9 LLM USAGE STATEMENT

We utilized Google's Gemini 2.5 Pro to improve the grammar and clarity of this manuscript. Drafts of several paragraphs were provided to the model to check for grammatical errors and suggest more natural phrasing. The authors reviewed all suggestions and retained full control over the final text.

