# OpenReview forum: "SparseCache: Extreme Sparse Coding for KV Cache Compression"
_ICLR.cc/2026/Conference — ICLR 2026 Conference Withdrawn Submission_

### Official Review · Reviewer_m8Gk · 2025-10-31

**Soundness:** 4
**Presentation:** 4
**Contribution:** 3
**Rating:** 8
**Confidence:** 3

**Summary:**

This paper addresses the problem of pre-computed RAG, where online prefill isnt needed but it increases storage requirements. To fix this, one can compress offline document KV-Caches into a sparse code and then reconstruct them on demand, this will drastically reduce the cost of storing these prefill-kvs and also make inference faster .This is done by learning a global dictionary for (K,V) each; shared across several head+layers and then producing per-token sparse codes, minimizing reconstruction loss by SGD.

**Strengths:**

- Paper addresses a very important problem, pre-computed RAG storage and improved time-to-first-token. Problem + solution is very clear, practical.
- Great empirical result on compression, exceeding 8x.
- Outperforms xkv, minicache etc, on compression ratio.
- clear TTFT breakdown is provided, very impressive improvements

**Weaknesses:**

Q2 What is the offline cost? at very large document costs, it can be very heavy. While real-world improvements will be seen at deployment, it might be nice to discuss it a bit more.
Q1 discussion on robustness of method when dictionary pre-training on different datasets and subsequent downstream results are not provided

**Questions:**

Regarding Q2, is this information missing because there is not sufficient diversity in downstream evaluation asks for document retrieval?

How is the position of the document handled, does this happen pre or post-rope? Do you have any comments on what would happen to retrieval if the ranking changes and the most-relevant doc is ‘very early’ in the context? This is not a deal-breaker for this paper, I am just curious about recency bias.

What would happen as document length increases from 512 tokens, do you expect achievable compression rates to fall as more tokens would have to be compressed? Is there any study on this?

---

### Official Review · Reviewer_Do1p · 2025-11-01

**Soundness:** 2
**Presentation:** 2
**Contribution:** 1
**Rating:** 2
**Confidence:** 4

**Summary:**

The motivation of this paper is to compress precomputed KV caches in a RAG scenario using sparse coding. However, the method and experiments do not consider a RAG scenario that involves multiple documents. When limited to standard (single-document) long-context KV cache compression scenarios, the method introduces only incremental design changes compared to prior KV cache compression methods based on sparse coding (https://arxiv.org/abs/2412.08890).

**Strengths:**

- Solid analysis of TTFT across different methods.

**Weaknesses:**

**[W1]** In the introduction and Figure 2, the paper discusses RAG, where multiple documents’ KV caches are precomputed following TurboRAG. However, the method itself does not address problems that arise from precomputing KV caches for multiple documents, such as overlapping position IDs and the inability to perform cross-document attention. The experiments also do not consider any RAG scenario in which multiple documents are given and their KV caches are precomputed separately.

**[W2]** Imprecise arguments:
- The paper title should state that it is about RAG.
- L120: This is unclear. How does the complexity overhead increase proportionally to model depth?
- L121: You argue that a large dictionary size can be a significant memory bottleneck for precomputed RAG systems. Why? A large dictionary does not necessarily imply a large compressed KV cache size. The memory overhead due to a large dictionary size is constant and does not scale with context length.
- In the introduction and methods sections, the paper states that the method learns a single unified dictionary across all layers, but in the experiments, it introduces \(L\), the number of consecutive layers sharing a dictionary.
- Dictionary learning: It is unclear what is meant by a “deep-learning-style training paradigm.” How is this training process different from Lexico (https://arxiv.org/abs/2412.08890)?

**Questions:**

Please see Weaknesses.

---

### Official Review · Reviewer_Rj9e · 2025-11-04

**Soundness:** 2
**Presentation:** 3
**Contribution:** 1
**Rating:** 2
**Confidence:** 5

**Summary:**

To deploy RAG with faster Time-To-First-Token (TTFT), one can precompute the context into KV cache beforehand, but this creates a huge storage requirement. The paper proposes SparseCache, a dictionary‑learning approach to KV‑cache compression aimed at making precomputed‑RAG practical. The method learns two global dictionaries, one for Keys and one for Values, shared across all layers and heads; the dictionaries are trained with an alternating scheme inspired by K‑SVD, but updated via mini‑batch gradient descent on a reconstruction loss. At preprocessing time, each KV vector is encoded via Orthogonal Matching Pursuit (OMP) into a sparse code of (indices, coefficients). At query time, the system reconstructs dense KV from those codes and feeds them to the model. The paper emphasizes a systems framing of an end‑to‑end RAG workflow, a clear method diagram, and algorithms for training/encoding/decoding.

Contributions:
- The paper reframes KV cache compression into the problem of precomputed RAG.
- On RULER (64k context) with Llama‑3.1‑8B, Qwen‑2.5‑7B/14B, SparseCache reportedly lies on the accuracy–compression Pareto frontier, maintaining strong accuracy at ~4–9× and remaining usable at 17.7× where baselines falter. It also reduces TTFT significantly versus both on‑the‑fly prefill and uncompressed precomputed caches.
- The paper uses 13‑bit index packing for dictionary atom IDs because 2^13=8192 matches the dictionary size, yielding an extra 1.23× on the indices.

**Strengths:**

**Originality:**
* Framing the contribution around precomputed‑RAG is interesting and useful. The paper integrates dictionary learning with an operational pipeline (offline compression → online reconstruction) and evaluates system‑level metrics (TTFT, memory).

**Quality:**
* Index bit‑packing (13‑bit) is a practical engineering detail that practitioners can replicate
* The experiments cover multiple LLMs and include some hyperparameter ablations over sparsity and layer‑sharing, plus systems metrics such as TTFT and memory.

**Clarity:**
* Figures effectively explain the workflow (Fig. 1) and sparse‑coding loop (Fig. 2).

**Significance:**
* If validated head‑to‑head, a “compress‑once, reuse‑often” cache for corpora could reduce TTFT and GPU memory use, directly addressing deployment bottlenecks for enterprise RAG. The paper’s focus on system‑level practicality is valuable.

**Weaknesses:**

In order of decreasing significance:

1. **Limited algorithmic novelty relative to Lexico:**
The core algorithmic method is a OMP‑based sparse coding of KV cache using universal dictionaries trained by alternating sparse coding and gradient updates, with codes stored as (indices, low‑precision coefficients. This is exactly the same as its cited work Lexico. There are only two novel things in this paper: one is a design choice (global dictionary across layers / head concatenation / index packing) and  the other is problem framing as precomputed‑RAG, rather than a new compression algorithm. Bit packing itself is too much of a standard optimization trick in quantization literature to call it a novelty.

2. **Missing direct baselines under equal conditions:**
Even though the work is directly inspired by the designs of Lexico, there are no Lexico baselines in any of their experiments. I expect Figure 3, 4a, 4b to all include Lexico for proper head-to-head evaluation. The paper is missing another line of important work, which is Cartridges (https://arxiv.org/abs/2506.06266). The method is not dictionary learning, but Cartrdiges paper is the most appropriate to cite and compare since they are concerned with how to compress a corpus of documents into the cache; in this aspect, they have advantages over Lexico in precomputed RAG usefulness.

3. **How is K-SVD alternating optimization different than that in Lexico?**
The term end-to-end signifies our approach of employing a deep-learning-style training paradigm using external text datasets to learn these dictionaries, rather than traditional Orthogonal Matching Pursuit (OMP)-based dictionary learning" (L84-86). The paper emphasizes their end-to-end design of dictionary learning. However, 1) SparseCache still uses OMP in its process so the above statement misleads the reader into thinking that no OMP is involved, and 2) how is SparseCache's training approach different than the approach in Lexico? Lexico did not express it as K-SVD, but it is essentially doing the same thing. Any pointers in the differences would be appreciated. Without a clear explanation in algorithmic differences, the K-SVD framing feels more like a decoy than an inspiration.

4. **Methodological coverage is narrow for RAG:**
Evaluation centers on RULER and synthetic long‑context probes. For a RAG‑motivated paper, there is no task‑level end‑to‑end evaluation (e.g., multi‑hop QA with real retrieval, domain shift scenarios), nor cross‑domain analyses for the learned dictionaries. The paper can be stronger if it can show that the dictionaries learned from one corpus can transfer to another corpus without re-training the dictionaries.

**Questions:**

1. How is K-SVD alternating optimization different than that in Lexico?
2. Can you add head‑to‑head results vs. Lexico on the same models/contexts under equal budgets (codes plus dictionaries)? These results are essential to assess originality and practical value.
3. Can you add task‑level RAG experiments (retrieval + generation) beyond RULER to show that TTFT and memory wins translate to end‑to‑end quality?
4. How do the dictionaries transfer to other domains? Can a single global dictionary generalize across model families or do you train per‑model per-task?
5. What is the relationship between reconstruction loss and downstream performance? How good does the reconstruction have to be and have you tested any other loss function than just the squared loss?

---

### Official Review · Reviewer_qWoQ · 2025-11-11

**Soundness:** 2
**Presentation:** 3
**Contribution:** 2
**Rating:** 2
**Confidence:** 4

**Summary:**

This article focuses on the problem of high memory usage and loading latency of KV cache in long-context / precomputed RAG scenarios. The authors propose SparseCache: a KV cache compression framework based on dictionary learning and sparse coding. The core points are:

1 For a given LLM, learn globally shared Key/Value dictionaries from offline collected KV vectors, rather than layer-specific or head-specific dictionaries.

2 Use OMP for sparse coding for each KV vector on the dictionary as a compressed representation; during online retrieval, reconstruct the KV from the sparse code and concatenate it with the query KV for inference.

3 For training, adopt an end-to-end dictionary learning approach to optimize reconstruction error; combined with engineering details such as index bit-packing to improve overall compression ratio.

4 Evaluated on Llama3.1-8B and Qwen2.5-7B/14B using the RULER long-context benchmark, as well as simulating TTFT and memory usage in TurboRAG. Results show that at the same level of precision loss, SparseCache maintains usable performance at higher compression rates compared with Minicache / Single SVD / xKV methods, and significantly reduces the KV storage and loading overhead for precomputed RAG2.

**Strengths:**

1 The problem is clearly defined with a focus on real pain points.

2 The method is somewhat innovative rather than just a simple accumulation of techniques.

3 The experimental coverage and analytical dimensions are relatively complete.

**Weaknesses:**

1 The term 'end-to-end' can easily cause misunderstanding or exaggeration. The current training objective is: obtaining codes using OMP, then updating the dictionary via gradient descent to minimize KV reconstruction error. Essentially, this is a deep learning implementation of K-SVD style alternating optimization; however, it does not actually incorporate downstream task loss or model logits into the objective, nor does it jointly fine-tune the LLM. It is recommended to clarify the terminology—for example, calling it 'end-to-end dictionary learning over KV activations with gradient-based updates' to avoid implying 'joint end-to-end optimization of LLM + dictionary.'

2 The comparison with the most relevant work is insufficient ,especially Lexico & quantitative SOTA methods. In the related work section, the paper mentions Lexico and several KV quantization / 1-bit / low-bit methods, but Table 2 only compares Minicache, Single SVD, and xKV, without directly running recent high-intensity baselines such as Lexico, KVQuant, CSR, or SVDQ.

3 Insufficient theoretical and convergence/stability analysis. Currently, only empirical results are provided, with no discussion on:
(1) The convergence of using OMP to generate codes, followed by gradient descent on the dictionary;
(2) Whether a globally shared dictionary has insufficient expressiveness or is 'over-shared' for different layers or attention heads, with theoretical or visualization analysis.
Although ICLR does not require rigorous theory, a bit could be added: for example, visualization of the learned atom distribution, inter-layer usage frequency, mutual information, or alignment with KV principal components, to support that 'the global dictionary indeed captures cross-layer redundancy.'

4 The experimental scenarios are relatively simple, lacking verification on real RAG/generation tasks.
The main evaluation focuses on the RULER and experiments simulating TTFT/memory.
Regarding claims about pre-computed RAG, there is a lack of experiments on end-to-end latency and accuracy in real retrieval QA or code assistant scenarios (such as multi-doc QA, tool-RAG, open-domain QA) and experiments in real systems under different numbers of retrieved documents and varying corpus scales.

5 Generality and transferability are not adequately explained. The dictionary is trained individually for each model and uses 3M tokens from Wikitext; it does not demonstrate whether the dictionary can maintain effectiveness across different domains/corpora and whether a dictionary learned on one model can be transferred to different fine-tuned versions of the same architecture, it also not show adaptability to tasks with large distribution differences, such as code or multi-modal tasks.

6 The system implementation comparison may not be entirely fair or transparent.TTFT comparisons depend on specific I/O bandwidth and implementation details. In the current setup, TurboRAG appears to be a naive version that simply loads uncompressed KV files, whereas actual systems can employ optimizations such as mmap, pipelining, and compression. It is recommended to clarify the baseline implementation assumptions more explicitly in the main text or appendix.

**Questions:**

1 Have you tried jointly optimizing the reconstruction error plus the downstream task loss? If so, what were the results and at what cost?

2 Besides the results for L1/L2/L4, is there any statistical or visual evidence showing that 'certain layers rely more on specific atoms,' thereby supporting the claim of 'global structural redundancy'?

3 In scenarios involving precomputations on extremely large corpora (>TB), would the cost of offline OMP compression become a new bottleneck? Have faster approximate sparse coders been evaluated?

4 Would further quantization of the sparse codes and the dictionary itself significantly affect performance? Are there any preliminary results?

---

### Note · Authors · 2026-01-27

**Comment:**

We acknowledge the constructive comments provided by the reviewers. We have decided to withdraw the manuscript to further improve the work and submit it to an alternative venue.

**Withdrawal Confirmation:**

I have read and agree with the venue's withdrawal policy on behalf of myself and my co-authors.

---

### Meta-Review · Area_Chair_KvNK · 2026-01-05

**Summary:**

This paper proposes SparseCache, a KV-cache compression approach based on dictionary learning + sparse coding (OMP) with globally shared dictionaries for Keys and Values, motivated by making “precomputed RAG” feasible by reducing KV storage and improving TTFT. The method trains dictionaries using an alternating procedure inspired by K-SVD (sparse coding + gradient-based dictionary updates) and includes practical engineering (e.g., index bit-packing). Experiments on long-context benchmarks (primarily RULER) and simulated TTFT/memory measurements suggest strong compression–accuracy trade-offs, including high compression ratios.

**Reviewer Concerns:**

Reviewers agree the problem is important and the paper is generally well written with useful systems framing and promising empirical results (one reviewer rates it as accept). However, three reviewers recommend reject, citing (i) insufficient novelty relative to prior sparse-coding KV compression (especially Lexico), (ii) missing head-to-head baselines against the most relevant methods, and (iii) a mismatch between the “precomputed RAG” framing and the evaluated/handled setting (limited or no end-to-end multi-document RAG evaluation and missing discussion of RAG-specific issues like cross-document attention / position handling). Several reviewers also flag unclear or potentially misleading wording around “end-to-end” optimization and ask for stronger analysis of transferability and offline preprocessing costs.

**Reviewer Scores:**

There is no author feedback submitted. Likely there will be no score changed.

---

### Decision · Program_Chairs · 2026-01-26

Reject